# META-LEARNING THE INDUCTIVE BIASES OF SIMPLE NEURAL CIRCUITS

## ABSTRACT

Animals receive noisy and incomplete information, from which we must learn how to react in novel situations. A fundamental problem is that training data is always finite, making it unclear how to generalise to unseen data. But, animals do react appropriately to unseen data, wielding Occam's razor to select a parsimonious explanation of the observations. How they do this is called their inductive bias, and it is implicitly built into the operation of animals' neural circuits. This relationship between an observed circuit and its inductive bias is a useful explanatory window for neuroscience, allowing design choices to be understood normatively. However, it is generally very difficult to map circuit structure to inductive bias. In this work we present a neural network tool to bridge this gap. The tool allows us to meta-learn the inductive bias of neural circuits by learning functions that a neural circuit finds easy to generalise, since easy-to-generalise functions are exactly those the circuit chooses to explain incomplete data. We show that in systems where the inductive bias is known analytically, i.e. linear and kernel regression, our tool recovers it. Then, we show it is able to flexibly extract inductive biases from differentiable circuits, including spiking neural networks. This illustrates the intended use of our tool: understanding the role of otherwise opaque pieces of neural functionality, such as non-linearities, learning rules, or connectomic data, through the inductive bias they induce.

## 1  INTRODUCTION

Generalising to unseen data is a fundamental problem for animals and machines: you receive a set of noisy training data, say an assignment of valence to the activity of a sensory neuron, and must fill in the gaps to predict valence from activity, Fig. 1A. This is hard since, without prior assumptions, it is completely underconstrained. Many explanations or hypotheses perfectly fit any dataset (Hume, 1748), but different choices will lead to wildly different outcomes. Further, the training data is likely noisy; how you choose to sift the signal from the noise can heavily influence generalisation, Fig. 1B.

Generalising requires prior assumptions about likely explanations of the data. For example, prior belief that small changes in activity lead to correspondingly small changes in valence would bias you towards smoother explanations, breaking the tie between options 1 and 2 in Fig. 1A. It is a learner's inductive bias that chooses certain, otherwise similarly well-fitting, explanations over others.

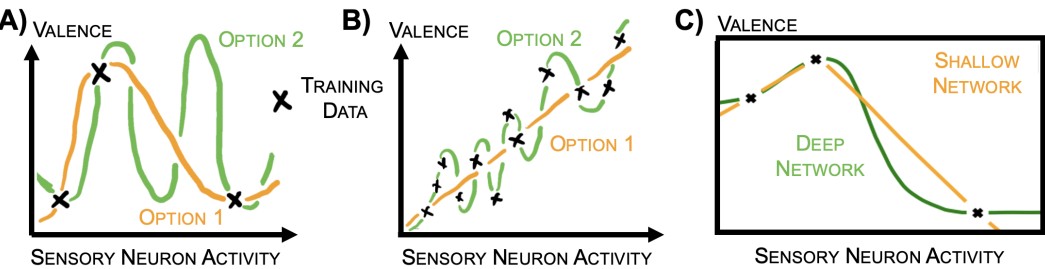

Figure 1: **Generalisation Requires Prior Assumptions. A:** The same dataset is perfectly fit by many functions. **B:** Different assumptions about signal quality lead to different fittings. **C:** Training a 2 (shallow) or 8 (deep) layer ReLU network on the same dataset leads to different generalisations.

The inductive bias of a learning algorithm, such as a neural network, can be a powerful route to understanding in both Machine Learning and Neuroscience. Classically, the success of convolutional neural networks can be attributed to their explicit inductive bias towards translation-invariant classifications (LeCun et al., 1998), and these ideas have since been very successfully extended to networks with a range of structural biases (Bronstein et al., 2021). Further, many network features have been linked to implicit regularisation of the network, such as the stochasticity of SGD (Mandt et al., 2017), parameter initialisation (Glorot & Bengio, 2010), early stopping (Hardt et al., 2016), or low rank biases of gradient descent Gunasekar et al. (2017).

In neuroscience, the inductive bias has been used to assign normative roles to representational or structural choices via their effect on generalisation. For example, the non-linearity in neural network models of the cerebellum has been shown to have a strong effect on the network's ability to generalise functions with different frequency content (Xie et al., 2022). Experimentally, these network properties vary across the cerebellum, hence this work suggests that each part of the cerebellum may be tuned to tasks with particular smoothness properties. This is exemplary of a spate of recent papers applying similar techniques to visual representations (Bordelon et al., 2020; Pandey et al., 2021), mechanosensory representations (Pandey et al., 2021), and olfaction (Harris, 2019).

Despite the potential of using inductive bias to understand neural circuits, the approach is limited, since mapping from learning algorithms to their inductive bias is highly non-trivial. Numerous circuit features (learning rules, architecture, non-linearities, etc.) influence generalisation. For example, training two simple ReLU networks of different depth to classify three data points leads to different generalisations for non-obvious reasons, Fig. 1C. In constrained cases analytic bridges have mapped learning algorithms to their inductive bias. In particular, the study of kernel regression, an algorithm that maps data points to a feature space in which linear regression to labels is then performed (Sollich, 1998; Bordelon et al., 2020; Simon et al., 2021), has been influential: all the cited examples of understanding in neuroscience via inductive bias have used this bridge. However, it severely limits the approach: most biological circuits cannot be well approximated as performing a fixed feature map then linearly regressing to labels!

Here, we develop a flexible neural network approach that is able to meta-learn the inductive bias of essentially any differentiable supervised learning algorithm. It follows a meta-learning framework (Vanschoren, 2019): an outer neural network (the meta-learner) assigns labels to a dataset, this labelled dataset is then used in the inner optimisation to train the inner neural network (the learner). The meta-learner is then trained on a meta-loss which measures the generalisation error of the learner to unseen data. Through gradient descent on the meta-loss, the meta-learner meta-learns to label data in a way that the learner finds easy to generalise. These easy-to-generalise functions form a description of the inductive bias. In other words, if the network receives a few training points from this function it will generalise appropriately, and generally the network will regularly use this function to explain finite datasets.

To our knowledge, the most related work is Li et al. (2021). Li et al. view sets of neural networks, trained or untrained, as a distribution over the mapping from input to labels. They fit this distribution by meta-learning the parameters of a gaussian process which assigns a label distribution to each input. This provides an interpretable summary of fixed sets of network. In our work we do something very different: rather than focusing on a fixed, static set of networks, we find the inductive biases of learning algorithms via meta-learning easily learnt functions.

In the following sections we describe our scheme, and validate it by comparing to the known inductive biases of linear and kernel regression. We then extend it in several ways. First, networks are inductively biased towards areas of function space, not single functions. Therefore we learn a set of orthogonal functions that a learner finds easy to generalise, providing a richer characterisation of the inductive bias. Second, we introduce a framework that asks how a given design choice (architecture, learning rule, non-linearity) effects the inductive bias. To do that, we assemble two networks that differ only by the design choice in question, then we meta-learn a function that one network finds much easier to generalise than the other. This can be used to explain why a particular circuit feature is present. We again validate both schemes against linear and kernel regression. Finally we show our tool's flexibility in a series of more adventurous examples: we validate it on a challenging differentiable learner (a spiking neural network); we show it works in high-dimensions by meta-learning MNIST labels; and we highlight its explanatory power for neuroscience by using it to normatively explain patterns in recent connectomic data via their inductive bias.

## 2    A NEURAL NETWORK TO META-LEARN INDUCTIVE BIASES

Our main contribution is a meta-learning framework for extracting the inductive bias of differentiable learning algorithms, Fig. 2A, that we describe in this section. In the outer-loop a neural network, the meta-learner, assigns labels to input sampled from some distribution, hence creating the real-world function that our circuit of interest will try to learn. The inner-loop learning algorithm, the learner, is the circuit whose inductive bias we want to extract; for example, a biological sensory processing circuit that assigns valences to inputs. When provided with a training dataset of inputs and labels the learner adjusts its parameters according to its internal learning rules. Then the generalisation error of the trained learner is measured on a held-out test set, and this is used as the meta-loss to train the meta-learner. This process repeats, retraining the learner at every iteration and iteratively developing the meta-learner's weights, until the meta-learner is labelling the data in a way that the learner finds easy to generalise after training on a few datapoints (we used around 30). Thus, the meta-learner has extracted a function towards which the learner is inductively biased.

As just outlined, the meta-learner will find the easiest-to-generalise function, usually the one that assigns all inputs the same label. To avoid this trivial function, we introduce a term in the meta-loss that forces the distribution of labels to take a particular (non-constant) form. Specifically, it measures the Sinkhorn divergence between the meta-learner's label distribution and a uniform distribution from $-1$ to $1$ (other divergences also work, Appendix B). The full pseudocode is in Algorithm 1.

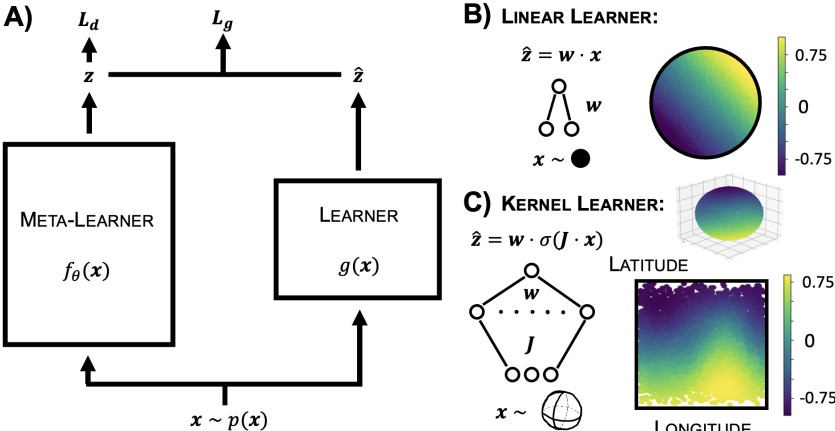

Figure 2: **Meta-Learning the Inductive Bias. A:** The meta-learner labels a dataset which is used to train the learner. Gradient descent is performed on a loss made of the learner's generalisation error to unseen data ($\mathcal{L}_g$), and the Sinkhorn divergence between the meta-learner's label distribution and a target distribution ($\mathcal{L}_d$), here choosen to be uniform from $-1$ to $1$. **B:** The meta-learner learns a linearly separable labelling of data sampled from a circle for a ridge regression learner. **C:** For a kernel regression learner and data sampled from the surface of a sphere, the meta-learner's labelling is very close to the predicted spherical harmonic (99% of norm within first order harmonics).

---

**Algorithm 1:** Pseudocode for Meta-Learning the Learner's Inductive Bias

---

1  Initialise meta-learner: $f_\theta(\boldsymbol{x})$
2  **while** *Step count < Total* **do**
3      Generate dataset from input distribution: $\boldsymbol{x} \sim p(\boldsymbol{x})$
4      Label using metalearner: $z = f_\theta(\boldsymbol{x})$
5      Split inputs and labels into test and train datasets: $\mathcal{D}_{Tr}$ & $\mathcal{D}_{Te}$
6      Train leaner using $\mathcal{D}_{Tr}$ giving trained learner network: $g(\boldsymbol{x})$
7      Label $\mathcal{D}_{Te}$ using trained learner: $\hat{z} = g(\boldsymbol{x})$
8      Compute the generalisation error of the leaner: $\mathcal{L}_g = \sum_i (z_i - \hat{z}_i)^2$
9      Compute the Sinkhorn Divergence of metalearner's labels from uniform $[-1, 1]$: $\mathcal{L}_d$
10     Take $\theta$ gradient step on meta-loss: $\mathcal{L} = \mathcal{L}_g + \mathcal{L}_d$
11 **end**

---

Our meta-learner must fit a function that the learner can generalise. To enable the meta-learner to learn all functions the learner might plausibly generalise well, its function class could usefully be a superset of the learner's. Therefore, we choose the meta-learner's architecture to be a slightly larger version of the learner's (though, beyond this, our findings appear robust, Appendix D).

We validate our scheme by meta-learning sensible functions for linear and kernel learners, whose inductive biases are known. First, for ridge regression on data sampled from a 2D circle the meta-learner assigns linearly separable labels, Fig. 2B; exactly the labels linear circuits easily generalise.

Next, we meta-learn kernel ridge regression's inductive bias. Kernel regression involves projecting the input data through a fixed mapping to a feature space (e.g. the last hidden layer of a fixed neural network) and performing linear regression from feature space to labels. Bordelon et al. (2020) show that the inductive bias of kernel regression can be understood through the kernel eigenfunctions ($\{v_i(\boldsymbol{x})\}$ with eigenvalue $\{\lambda_i\}$). These are defined on input distribution $p(\boldsymbol{x})$ via a kernel $k(\boldsymbol{x}, \boldsymbol{x}')$ that measures the similarity of two inputs in feature space:

$$\int k(\boldsymbol{x}, \boldsymbol{x}')v_i(\boldsymbol{x}')dp(\boldsymbol{x}') = \lambda_i v_i(\boldsymbol{x}). \tag{1}$$

The algorithm is inductively biased towards higher eigenvalue eigenfunctions; i.e., fewer training points are needed to reach a given generalisation error when fitting high vs. low eigenvalue eigenfunctions. General functions can be understood by projecting onto the eigenbasis. Hence our meta-learner, in searching for kernel regression's easiest-to-generalise non-constant function, should choose the highest eigenvalue eigenfunction.

To test this, we meta-learn the inductive bias of a two-layer neural network with fixed first layer weights. We sample data uniformly from the sphere and randomly connect a large hidden layer of ReLU neurons to the three input neurons. The elements of this random weight matrix are drawn *iid* from a standard normal, and the learning algorithm performs ridge regression on the hidden layer activities. Previous work has analytically derived the kernel for this network, and computed it's eigenfunctions (Cho & Saul, 2009; Mairal & Vert, 2018), which are spherical harmonics. The higher the frequency of the spherical harmonics the lower its eigenvalue. Matching this, our network meta-learns one of the set of lowest frequency spherical harmonics, Fig. 2C.

## 3 META-LEARNING AREAS OF FUNCTION SPACE

Having validated our tool on some simple test cases, we now extend it to find a richer characterisation of the inductive bias. A given learning algorithm is inductively biased towards areas of function space, not just one particular function. To gain access to this larger space, we learn a series of meta-learners. The first of these is exactly as described above, then we iteratively introduce additional meta-learners. To ensure each meta-learner learns a new aspect of the inductive bias we add a term to the meta-loss that penalises the square of the dot product between the current meta-learner's labelling and that of all the previously trained meta-learners, Fig. 3A. On a dataset $\{\boldsymbol{x}_n\}$:

$$\mathcal{L}_{\text{Orthog}} = \sum_i \left( \sum_n f_{\theta_i}(\boldsymbol{x}_n) f_{\theta'}(\boldsymbol{x}_n) \right)^2 \tag{2}$$

for each previous meta-learners $f_{\theta_i}(\boldsymbol{x})$ and the current meta-learner $f_{\theta'}(\boldsymbol{x})$. From the learner's perspective nothing has changed, at each meta-step it simply learns to fit the meta-learner that is currently being trained. But each additional meta-learner must discover an easy-to-generalise function that is orthogonal to all previous meta-learners.

We again validate this scheme against linear and kernel regression. When tested on linear regression of 2D data the meta-learners learn two orthogonal linearly separable labellings, then a third orthogonal function that the learner struggles to generalise, as expected, Fig. 3B. We then test on the same kernel regression network we described previously. Theory predicts that the meta-learners should learn the eigenfunctions in decreasing order of their eigenvalue, and we find that this is true to a good approximation, Fig. 3C, learning approximations to the three first order spherical harmonics, and then three approximations to second order spherical harmonics.

For linear classifiers (e.g. linear and kernel regression), the full set of orthogonal functions explains the entire inductive bias. This won't be true in general. Nonetheless, we expect the set of orthogonal functions will still be a helpful guide to a network's inductive bias, even for non-linear classifiers.

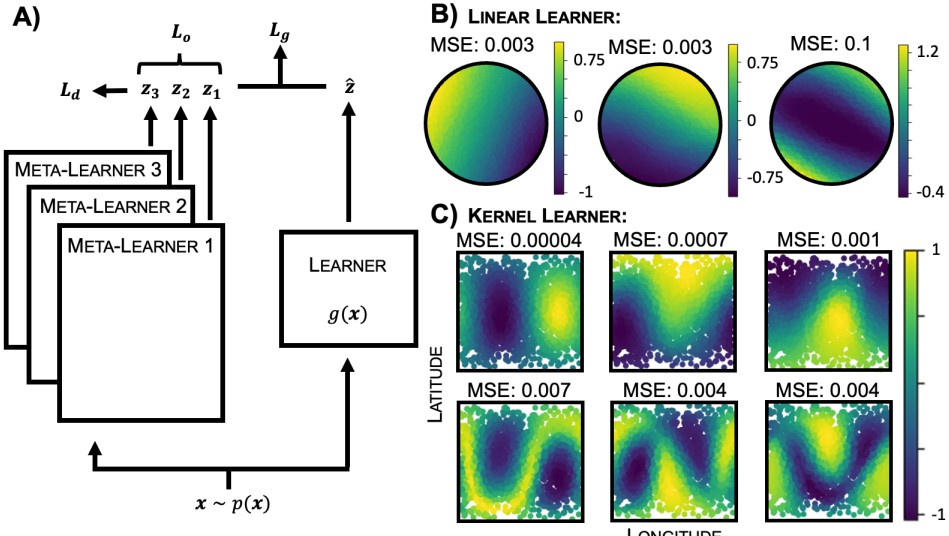

Figure 3: **Meta-Learning Many Functions. A:** We learn many meta-learners, each of which has to label orthogonally to all previous meta-learners. **B:** For a linear learner the meta-learners learn two orthogonal linear functions and an orthogonal but hard to learn third function. **C:** For a kernel learner we learn 6 meta-learners, the first 3 approximate well first order spherical harmonics (96% norm overlap), and the next 3 second order spherical harmonics (91% norm overlap), as predicted.

## 4 FINDING THE EFFECT OF DESIGN CHOICES ON THE INDUCTIVE BIAS

Our work is motivated by the desire to understand how design choices in learning algorithms - such as architecture, learning rule, and non-linearities - lead to downstream generalisation effects, particularly in biological networks. One additional setting which we have found useful is to compare two networks with some architectural difference between them, and learn functions that one of the networks finds much easier to generalise than the other. In this way, we can build intuition for the impact of design features on the inductive bias. To illustrate this we again create a meta-learner that labels data, but this time the labels are used to train two learners. We then train the meta-learner so that one learner (the chosen student) is much better at generalising than the other (neglected student). This is done by minimising the generalisation errors of the chosen student minus the neglected student, Fig. 4A. Validating this approach on well understood algorithms, we show that it can find functions that a kernel regression algorithm is able to learn better than linear regression, Fig. 4B, i.e. a non-linearly separable function.

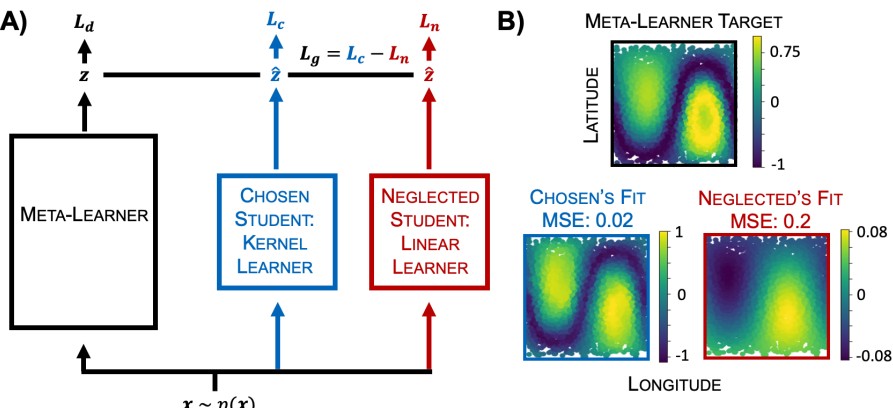

Figure 4: **Meta-learning design choice impact. A:** Labellings are learnt such that a chosen student generalises much better than a neglected one. **B:** The meta-learner finds a non-linear labelling for which kernel regression generalises an order of magnitude better than linear regression.

This illustrates some of the games that can be played in this setting. For example, you could play a co-operative game, in which you meta-learn a function that a set of learners all find easy to generalise, and each learner could have different connectivity matrices to match the distribution in real animals, ensuring the tool does not over-fit to some specific details. However as the losses become more complex training becomes harder, for example this adversarial setting between chosen and neglected student is hard to make robust if the two learners are relatively similar.

## 5 META-LEARNING APPLIED TO MORE COMPLEX LEARNING ALGORITHMS

So far we have developed and tested a suite of tools for extracting the inductive bias of learning algorithms. We now apply our tools to networks whose inductive bias cannot be understood analytically. Specifically: we show our method works on a challenging differentiable learner, a spiking neural network; we validate our method on a high-dimensional MNIST example; and we illustrate how our tool can give normative explanations for biological circuit features, by meta-learning the impact of connectivity structures on the generalisation of a model of the fly mushroom body. Our tool is flexible: by taking gradients through the training procedure we can meta-learn inductive biases for networks trained using PyTorch. We will provide code on github that produces our figures, including a basic ReLU network (Appendix A) which should be easily adapted to networks of interest.

### 5.1 SPIKING NEURAL NETWORK

The brain, unlike artificial neural networks, computes using spikes. How is an open question. A recent exciting advance in this area is the surrogate gradient method, which permits gradient based training of spiking neural networks by smoothing the discontinuous gradient (Neftci et al., 2019; Zenke & Vogels, 2021). We make use of this development to meta-learn the inductive bias of a spiking network, providing a challenging test case for our method.

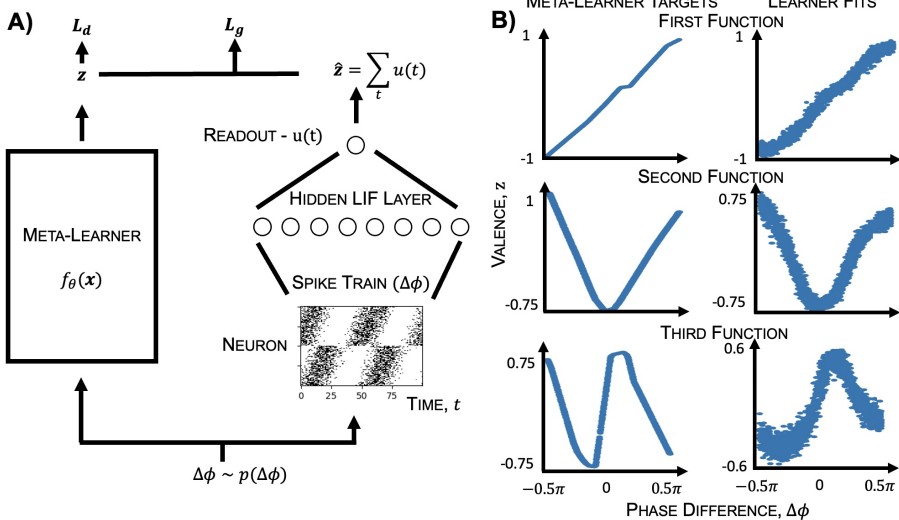

Figure 5: **Meta-learning through a Spiking Network. A:** Labellings are learnt that the spiking network, with weights trained via surrogate gradient descent, finds easy to generalise. Phase differences, $\Delta\phi$, are sampled uniformly and used to generate spike train by sampling from a poisson process with the following rates: for half the neurons $r_n = \frac{r_{max}}{2}(1 + \sin(t + \theta_n))^2$, where $n$ is a neuron index and $\theta_n$ are uniformly sampled offsets; for the other half we add a phase shift: $r_n = \frac{r_{max}}{2}(1 + \sin(t + \theta_n + \Delta\phi))^2$. These populations represent sensory neurons in the two ears, and $\Delta\phi$ is the interaural phase difference. This activity feeds into a population of linear-integrate-and-fire neurons, then onwards to a readout linear-integrate neuron. The valence assigned is the sum of the readout's activity over time. **B:** We learn three orthogonal meta-learners (as in section 3) and find the spiking network finds it easiest to learn low frequency functions. Left: the meta-learner's target function. Right: the spiking network's labelling. As can be seen, the spiking network captures the main behaviour, but increasingly poorly at higher frequencies.

We study a modification of a model developed for a recent tutorial (Goodman et al., 2022; Zenke, 2019), which is trained to assign a label to an incoming spike train. The network is a model of an interaural phase difference detection circuit. The input spike train is parameterised by a phase difference, $\Delta\phi$, that generates two sets of spike trains, one in each ear, Fig. 5A. These spikes are processed through a hidden layer of linear-integrate-and-fire neurons (LIF), before reaching a classification layer. A real-valued valence is assigned by summing the output neuron's activity over the trial. The meta-learning framework is as before: the meta-learner assigns valences to input phase differences, these labels are used to train the spiking network by surrogate gradient descent, then the meta-learner is trained to minimise the learner's generalisation error and a distribution loss. Our method works well, finding a simple smoothness prior, Fig. 5B.

## 5.2 A High-Dimensional MNIST Example

Next, we test out method on a high-dimensional input dataset. Thus far, to visualise our results, we have only considered low dimensional input data. We demonstrate that our method continues to work in high-dimensions by applying it to a dataset made of the 0 and 1 MNIST digits (LeCun, 1998). We meta-learn a labelling of this dataset that a simple convolutional neural network finds easy to generalise. Our meta-learner's architecture is also a convolutional neural network whose outputs are bounded between 0 and 1, and the meta-learner must learn an easy-to-generalise labelling with high variance. We find that the meta-learner consistently rediscovers the MNIST digits within the dataset, separating each digit into its own class, figure 6. We return to the important question of understanding high-dimensional inductive biases in the discussion.

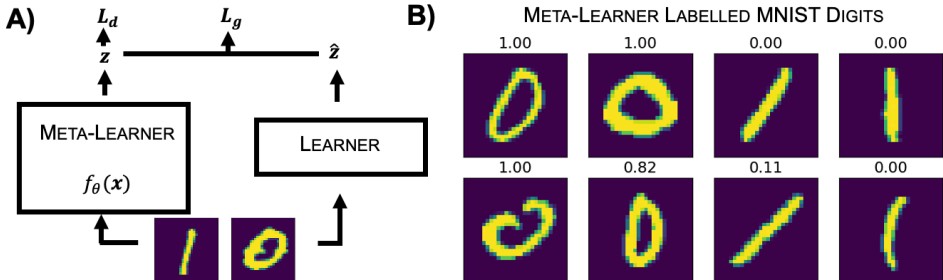

Figure 6: **Meta-Learning on MNIST A:** A meta-learner receives MNIST 0s and 1s, and assigns labels, bounded between 0 and 1, that have high variance and can be easily generalised by the learner. **B:** 99% of digits are assigned a label, shown in title, consistent with MNIST class.

## 5.3 Interpreting Connectivity Patterns through their Induced Inductive Bias

A large maturing source of neuroscience data is (a list of which neurons connect to one another). However, there is currently a dearth of methods for interpreting this data (Litwin-Kumar & Turaga, 2019). In this section, we show our tool can be used to give normative roles to connectomic patterns through their induced inductive bias. We study a model of the fly mushroom body, a beautiful circuit that fruit flies use to assign valence to odours (Aso et al., 2014; Hige, 2018), for which connectomic data has recently become available (Zheng et al., 2018; 2022).

Odorants trigger a subset of the fly's olfactory receptors. These activations are represented in a small glomerular population (input neurons), projected to a large layer of Kenyon cells (hidden neurons), then onwards to output neurons that signal various dimensions of the odour's valence, Fig. 7A. An error signal is provided if the fly misclassifies a good odour as bad, or vice versa, allowing the fly to update its weights and learn appropriate responses. Classically, the input-to-hidden connectivity was assumed random; i.e., each hidden neuron connects to a few randomly selected input neurons. However, connectomic data has shown that hidden neurons preferentially connect to some inputs, and there are input groupings - if a hidden neuron connects to one member of a group it likely connects to many, Fig 7D (Zheng et al., 2018; 2022). Zavitz et al. (2021) tested networks with this connectivity on a battery of tasks and found that, compared to random, (1) they were better at identifying odours that activated over-connected inputs, and (2) they generalised assigned valence across a group (i.e. if you assign high valence to the activation of one neuron, you do the same for other neurons in the same group).

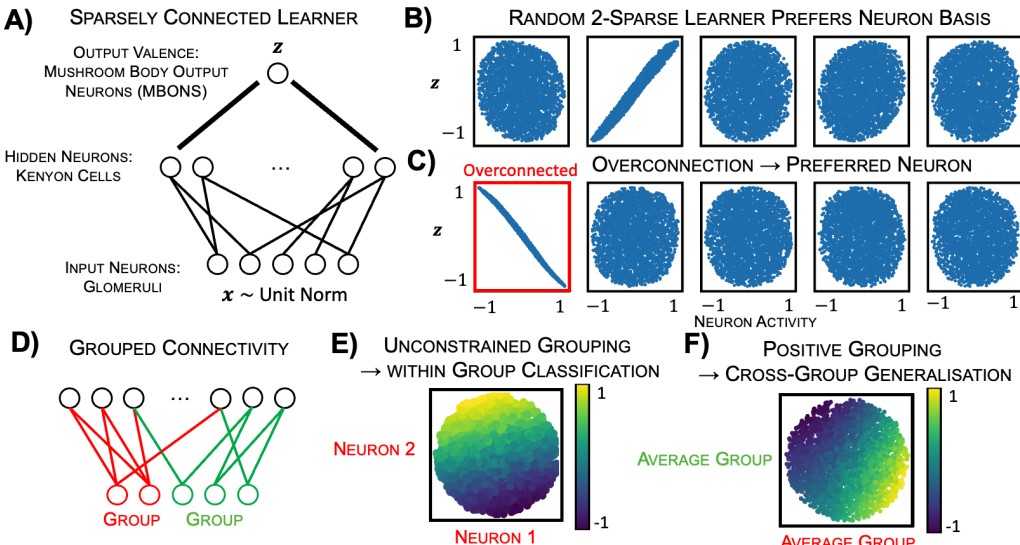

Figure 7: **Understanding Connectivity via Inductive Bias. A:** We model the fly mushroom body as a ReLU network with one large hidden layer. Each hidden neuron is connected to two of the five input neurons. **B:** The meta-learner finds the labelling the learner generalises most easily. We show this labelling projected against each of the input neuron activities. As can be seen, the labelling depends on only one neuron's activity, second from left. **C:** In the overconnected setting each hidden neuron still connects to two inputs, but there is a strong bias towards connecting to the first, highlighted neuron. As a result, the meta-learner settles on a labelling that depends only on this neuron's activity. **D:** We explore the impacts of group connectivity, in which the input neurons are divided into two groups, and hidden neurons tend to be connected to two neurons from the same group. **E:** We train the meta-learner, and find that it's labelling depends only on neurons within the same group. The plot shows the projection of the datapoints into a subspace defined by the two neurons in the red group. The labelling depends linearly on position within this subspace. **F:** However, if the input-hidden connections are constrained to be positive, the meta-learner's labelling depends only on the average activity within each group, i.e. if one member of a group increases the output, so do all members; hence, the function generalises across group members.

We used our tool to verify and develop these findings by examining the effect of different connectivity patterns on the inductive bias of a sparsely-connected model of the mushroom body, Fig. 7A. As a baseline, fully connected networks are biased towards smooth functions, appendix A, the simplest being those that assign valence based on one direction in the input space: high at one end, low at the other, like in Fig. 2B - C. However, which direction is unimportant; they're all equally easy to learn. Sparsity breaks this degeneracy, aligning the easiest to learn functions with the input neuron basis, figure 7B. As such, sparse connectivity, which is ubiquitous in neuroscience, ensures the fly is best at assigning labels based on the activity of small collections of neurons. Next, we introduced the observed connectomic structure. Biasing the connectivity broke the degeneracy amongst neuron axes. The networks were, fairly intuitively, best at generalising functions that depended on the activity of overconnected inputs, figure 7C, matching Zavitz et al. (2021). Finally we introduce connectivity groups, figure 7D. Without additional changes this does little, the neuron basis is still preferred and, unlike Zavitz et al. (2021), generalisation across inputs is not observed, figure 7E. Only when we additionally constrain the input-to-hidden connections to be excitatory (i.e. positive) do we see that the circuit becomes inductively biased towards functions that generalise across groups of inputs, figure 7F. In retrospect this can be understood intuitively: positive weights and grouped connectivity ensure that a hidden neuron that is activated by one input will also be activated by other group members, encouraging generalisation. This effect is removed by permitting negative weights, which let members of the same group excite or inhibit the same hidden neuron.

Thus, we verify the findings of Zavitz et al. (2021) without needing to presuppose a battery of tasks. In doing so we highlight how our tool can be used to gain insight into the role of circuit design choices, in particular, the importance of the neuron basis for sparsely connected networks.

# 6   DISCUSSION & CONCLUSIONS

We presented a meta-learning approach to extract the inductive bias of differentiable supervised learning algorithms, which we hope will be useful in normatively interpreting the role of features of biological networks. This approach required few assumptions beyond those that make the inductive bias an interesting way to conceptualise a circuit in the first place. We required, first, the circuit must be interpretable as performing supervised learning. Second, the input data must be specified. And, third, you must specify the way the circuit learns, and be able to take gradients through this learning process. We will discuss each of these requirements and ways they could be relaxed; regardless, it is heartening that any circuit satisfying these will, in principle, suffice. The analytic bridge between kernel regression and its inductive bias (Bordelon et al., 2020; Simon et al., 2021) has already found multiple uses in biology in just a few years (Bordelon & Pehlevan, 2021; Pandey et al., 2021; Harris, 2019; Xie et al., 2022), despite its stringent assumptions. We hope that relaxing those assumptions will offer a route to allow these ideas to be applied more broadly.

The first requirement is that the learner performs supervised learning. This is often reasonable. Some circuits contain explicit supervision or error signals, like the fly mushroom body or the cerebellum (Shadmehr, 2020), and generally brain areas that make predictions (i.e., all internal models), can use their prediction errors as a learning signal. Alternatively, some circuits are well modelled as one area providing a supervisory signal for another, as in classic systems consolidation (McClelland et al., 1995), or receiving supervision from a past version of themselves through replay (van de Ven et al., 2020). Nevertheless, modelling circuits as performing supervised learning will always be an approximation, most simply due to unmodelled effects such as neuromodulation. As an illustration of how our framework could be extended beyond supervised learning, we consider how neuromodulation could be incorporated. There exist models of how neuromodulation influences circuit function, for example by sharpening neural non-linearities (Ferguson & Cardin, 2020; Aston-Jones & Cohen, 2005), and these could be included in the learner model. The meta-learner could then meta-learn two outputs, one label and one neuromodulator. It's goal would be to meta-learn functions that the learner finds easy to generalise with limited quantities of neuromodulatory attention applied to specific, exemplar, training points.

Next, we could relax our second assumption, access to an input distribution, which is often lacking. This can be avoided by using real neural data as the input. Or, if neural data is limited, generative modelling could be used to fit the neural data distribution and new samples drawn from that distribution. Finally, one could imagine a single meta-learner that creates not only the label, but also the data. That is, the meta-learner could generate the entire dataset by transforming a noise sample into an input-output pair. This would have to be carefully regularised to avoid trivial input distributions, but could in principle learn the input statistics that particular networks are tuned to process.

Finally, in a slightly kooky way, we could avoid specifying the learning algorithm by interfacing with an animal directly! Animals' inductive biases are objects of interest in their own right, but can also give insight into the underlying neural processing. These biases could be studied by replacing the inner learner with a real animal that is trained on a labelled dataset from the meta-learner, then tested on new datapoints. Since we cannot compute gradients through the computations of a living animal, the meta-learner could be optimised using black-box optimisation procedures that rely only on meta-loss evaluations, like the Nelder-Mead method (Singer & Nelder, 2009).

Despite our optimism for this approach, there remain challenges. Most fundamentally, sets of functions that a learner easily generalises are still hard to interpret. We have shown how our tool can provide insight for low-dimensional inputs (Fig. 2 - 5), by comparing to ground truth labels (Fig. 6), or by projecting the learnt functions onto an appropriate basis (Fig. 7). However, to make the concept of inductive bias more powerful, more tools are needed to interpret the resulting functions.

To conclude, the inductive bias is a promising angle from which to understand learning algorithms. Analytic bridges between circuit design and inductive bias have already 'explained' the presence of aspects of the circuit through their effect on the network's generalisation properties in both artificial (Canatar et al., 2021; Bahri et al., 2021) and biological (Bordelon & Pehlevan, 2021; Pandey et al., 2021; Harris, 2019; Xie et al., 2022) networks. However, these techniques require very constraining assumptions. We have dramatically loosened these assumptions and shown our meta-learning approach can flexibly extract the inductive bias of neural circuits. We have shown its utility in interpreting connectomic data, and we believe it will prove useful on other datasets and problems.

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

## A   META-LEARNING A SIMPLE BACKPROP TRAINED ReLU NETWORK

A simple test of our framework is a feedfoward ReLU network with 2 hidden layers, learnt using gradient descent. While the functions this network finds easy to generalise cannot be extracted analytically, our tool finds that, unsurprisingly, these networks are biased towards smooth explanations of the data, learning six smooth orthogonal classifications that increase in frequency and mean squared error 8B. We include this example as the simplest PyTorch implementation of learning the inductive bias of a network trained by gradient descent, in the hope that the code can be easily adapted for future use.

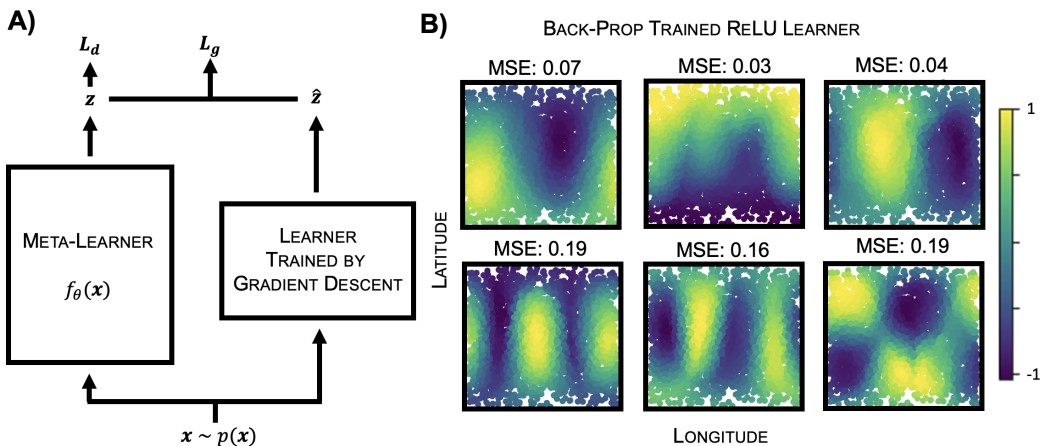

Figure 8: **Meta-learning a Backprop Trained Network A: Schematic** We meta-learn the functions a 2-layer ReLU network trained using backprop finds easy to generalise. **B: Low Frequency Bias**

## B   USING DIFFERENT DIVERGENCE MEASURES

To persuade the meta-learner to find non-trivial functions, we include a divergence loss that forces the meta-learner's label distribution to take a particular form: uniform between -1 and 1. In this section we show that the particular divergence that we use has little impact on the solutions we find for learning the easiest-to-generalise function of the kernel learner, figure 2C. Figure 9 shows that a variety of divergence metrics can be used, sinkhorn, an energy statistic from Székely & Rizzo (2013), and the maximum mean discrepancy from Gretton et al. (2012) (implementations from Djolonga (2020)). In each case the learnt function has over 99% norm in the space of first order spherical harmonics, demonstrating that the meta-learner has learnt appropriately.

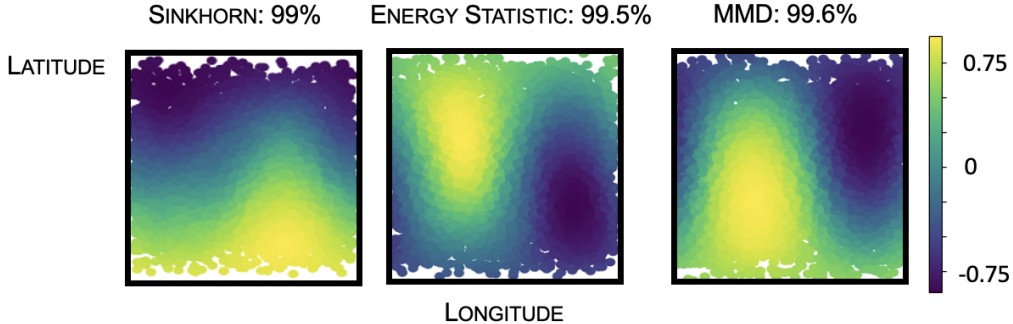

Figure 9: We use three different divergence metrics to penalise the meta-learner's label distribution for deviating from uniform between -1 and 1. We add these to the meta-loss, along with the generalisation error of the simple kernel learner introduced in section 2. For all three divergence metrics the meta-learner learns a very close approximation to a first order spherical harmonic, as predicted.

## C  RANDOM RE-SEEDING CANNOT REPLACE ORTHOGONALISATION

We introduced the orthogonalisation procedure in section 3 so that successive meta-learners have to explore larger areas of function space. It is legitimate to wonder whether simply re-running the optimisation would have had the same effect. Here we show it does not, and orthogonalisation is needed to explore the learner's inductive bias fully.

We re-run the meta-learner optimisation without any orthogonality terms for the kernel learner described in section 2. In figure 10 we find that the meta-learners find different functions, but only approximations to the first order spherical harmonics. This makes sense, the meta-learner is tasked with finding the easiest-to-generalise non-constant function. For this particular learner there is a degenerate space of such functions and so re-running the meta-learner simply draws another sample from this space of functions. However, to access the second order spherical harmonics that this learner still learns, just worse, we need something like the orthogonality constraint.

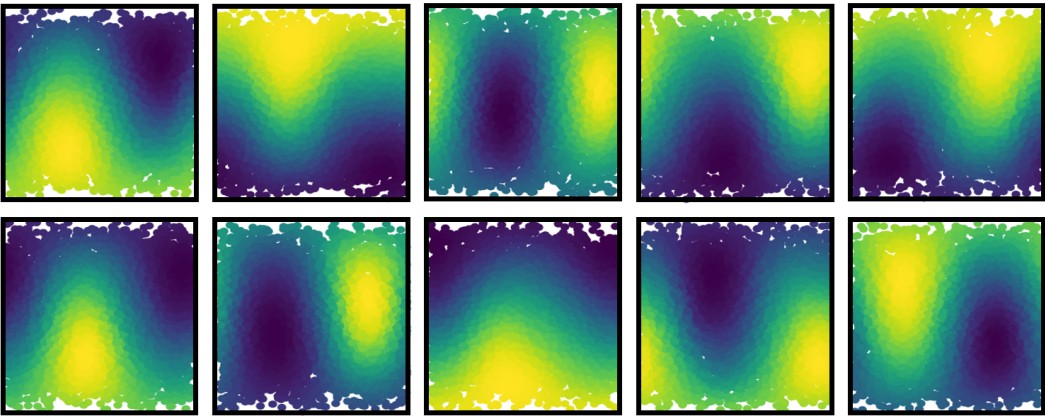

Figure 10: Re-running the meta-learner 10 times on the kernel regression algorithm from section 2 produces approximations to the first order spherical harmonics, but no second order functions.

## D  IMPACT OF META-LEARNER ARCHITECTURE ON EXTRACTED FUNCTIONS

We chose the meta-learner's architecture to be a slightly larger version of the learner's. Our motivation for this is that we want the function class of the meta-learner to be a super-set of the learner's, so that it can learn all the functions the learner could plausibly generalise well.

We tested how robust our results were to architectural changes in the meta-learner. We used the simple feedforward 2-hidden layer ReLU network as in Appendix A, trained by backpropagation, and learnt 6 orthogonal functions that the learner finds easy to generalise. We took our meta-learner to be a similar feedforward network of different depth from 4 to 0 hidden layers. Figure 11 shows that the specific choice of meta-learner didn't matter for meta-learners with between 2 to 4 layers. Each learnt 3 approximations to first order spherical harmonics, and 3 to second order harmonics. But a linear learner can't learn more than three orthogonal functions, so failed to find more than three orthogonal generalisable functions.

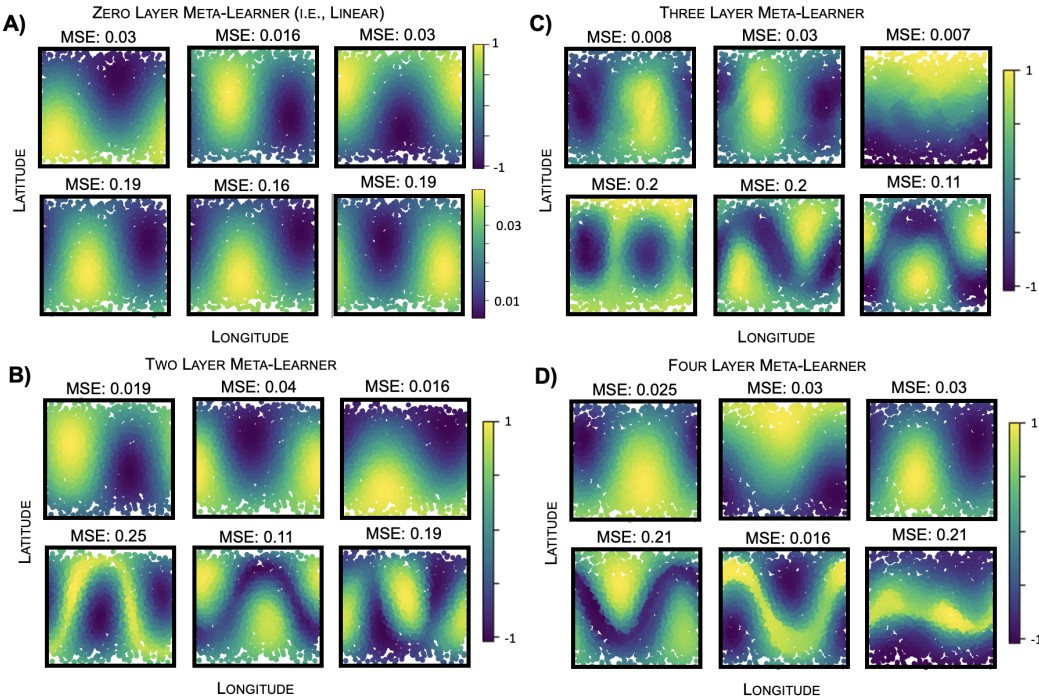

Figure 11: We meta-learn 6 easy-to-generalise functions for a simple ReLU network using meta-learners of different depths. This process fails to find more than 3 orthogonal functions when the meta-learner is linear (A.) (note the small label spread, a symptom of a failure to learn), but meta-learners with (B.) 2, (C.) 3, or (D.) all find qualitatively similar results

