# OpenReview forum: "Meta-Learning the Inductive Biases of Simple Neural Circuits"
_ICLR.cc/2023/Conference — Submitted to ICLR 2023_

### Official Review · Reviewer_TZjT · 2022-10-24

**Confidence:** 3
**Correctness:** 4
**Technical Novelty And Significance:** 3
**Empirical Novelty And Significance:** 4
**Recommendation:** 8

**Clarity, Quality, Novelty And Reproducibility:**

- generally the writing is clear and easy to follow

**Strength And Weaknesses:**

+ conceptually simple procedure, easy to apply to a range of models
+ sequential introduction of multiple meta-learners, intriguing multi-learner scenarios for comparative across models analysis
+ code provided for all demos

- there is a little bit of circularity in that the set of functional hypotheses being considered itself comes from a functional class with its own inductive biases.
- would have liked a little more details about the multi meta-learner scenario from the perspective of the inner loop learner

**Summary Of The Paper:**

The goal of the paper is to introduce a numerical procedure for extracting inductive biases, defined as ease of generalization,  for a differentiable trained recognition model. This procedure involves two layers of optimization: an inner learning model optimized on a training set of input-output pairs, and a meta-learner which is optimized so as to pick the training set that makes it easier for the inner learner to generalize to unseen data.

**Summary Of The Review:**

I like this a lot, found the approach clever and elegant. I imagine it would make a fun contributed talk/spotlight

---

> ### Author Response · Authors · 2022-11-18
> **Thanks for the Review! Response:**
>
> We thank you for spending time reading our work. We appreciate your comments and your complimentary review of our work. Below, we try to address the weaknesses highlighted and we hope the manuscript, updated in response to comments, makes things clearer. We look forward to answering any further questions you have in the coming weeks.
>
> > **Weakness 1:** there is a little bit of circularity in that the set of functional hypotheses being considered itself comes from a functional class with its own inductive biases.
>
> This is an insightful comment, indeed, the meta-learner is also a network with its own function class and inductive bias. We tend to choose the meta-learner’s architecture to be a larger version of the learner’s, the motivation being that we want the space of learnable functions for the meta-learner to be a more expressive version of the learner’s, since the meta-learner only needs to learn to fit functions that the learner can generalise. As such, this circularity is actually quite helpful to us (parameterising the function with a simple grid of values doesn’t work, for example, even in low-dimensional input spaces). For the set of meta-learners we have chosen (using the heuristics mentioned) we have got consistent results. Of course, not all choices work: making the meta-learner linear, for example, would not work as it precludes learning non-linear functions.
>
> > **Weakness 2:** would have liked a little more details about the multi meta-learner scenario from the perspective of the inner loop learner
>
> Apologies for the lack of clarity. At every meta-step the learner is reinitialised and trained. As such, in the multi-meta-learner scenario life continues as normal for the learner. It is reinitialised and trained at every meta-step. The only things that change are which meta-learner labels the training data, and the shape of that meta-learner’s loss, which includes extra terms as appropriate. Do you think this should be clarified in the main text?

---

> > ### Comment · Reviewer_TZjT · 2022-11-18
> > **Response**
> >
> > Thanks for the clarifications. It would not hurt for these to make their way to the actual text if there's room.

---

> > > ### Author Response · Authors · 2022-11-18
> > > **Clarifications added to main paper**
> > >
> > > Fair enough!
> > >
> > > We've added the following clarification of the meta-learner's architecture to section 2:
> > >
> > > *Our meta-learner must fit a function that the learner can generalise. To enable the meta-learner to learn all functions the learner might plausibly generalise well, its function class could usefully be a superset of the learner's. Therefore, we choose the meta-learner's architecture to be a slightly larger version of the learner's (though, beyond this, our findings appear robust, Appendix D).*
> > >
> > > (And a small appendix D that shows we get the same effects for multiple meta-learners with different architectures)
> > >
> > > And we've added this clarification to section 3 about the multi-meta-learner setup from the learner's point of view:
> > >
> > > *From the learner's perspective nothing has changed, at each meta-step it simply learns to fit the meta-learner that is currently being trained.*

---

### Official Review · Reviewer_wT2V · 2022-10-24

**Confidence:** 4
**Correctness:** 3
**Technical Novelty And Significance:** 2
**Empirical Novelty And Significance:** 2
**Recommendation:** 5

**Clarity, Quality, Novelty And Reproducibility:**

The paper is clearly explained, but some of the claims of the usefulness of the approach are lacking as mostly known results are recovered, and little to no predictions are made that would have to be validated. The authors rightfully point out that some tools are lacking to properly study their model.

**Strength And Weaknesses:**

The paper is well-written, and the breadth of models studied appears sufficient to touch a broad audience of ML and Neuroscience enthusiasts. The authors offer an alternative to the GP (Li et al.) and theoretical (Bordelon et al.) approaches.


There are many weaknesses with the current approach, which I believe makes it a bit premature for publication.

1) The choice of the Sinkhorn distance seems somewhat arbitrary. How does the selection of the different objectives affect learning?
2) The paper offers primarily qualitative comparisons (i.e., visual) rather than quantitative ones. Are there ways to properly understand the quality of the approximation of the implicit by different types of meta-learners?
3) How does the meta-learner's choice affect the implicit bias (given the meta-learners have their own inductive biases)?
4) What type of novel, non-trivial, insights can we obtain from the current paper? Various future directions and discussions are nicely laid out. However, as of now without them, this paper does not offer enough to be even a proof of concept.



**Summary Of The Paper:**

## Updated Score. Changed from 3 to 5

This paper proposes to address the critical question of understanding the inductive bias of machine learning models, i.e., what kind of function/problem models are most suited based on their design choices or other parameters like learning rules, etc.

The authors choose to design a model that can be used to characterize the inductive biases of various models systematically. The authors show that the meta-learner for given tasks and simple, relatively well-understood models can learn the inductive bias functions that the learning model itself has. They extend their study to more complex "standard" neural networks trained with backprop and spiking neural networks.

Although the paper offers an interesting approach, one important objection is that, as opposed to the paper from Li et al. 2021, the meta-learners would themselves require a meta-learner to understand their own biases.



**Summary Of The Review:**

The paper addresses an interesting question for both the ML and Neuroscience community. Given an observed model, how can we determine its inductive bias when its theoretical study is hard or impossible?

The paper, nonetheless, does not address critical questions and make enough predictions to make this paper even a proof of concept. It appears novel because it uses neural networks instead of GPs as meta-learners. Still, the resulting interpretability is lost because the meta-learners inductive bias is hard to understand.

I look forward to hearing from the authors and clarifying points I might have missed.

---

> ### Author Response · Authors · 2022-11-18
> **Thanks for the Review! Response (1/3)**
>
> Thank you for your attentive and helpful reading of our work. We have tried to respond to each of your comments, and have included new text and figures in the manuscript that help us to address points you have raised. We look forward to further discussion of any aspects that remain unclear, wrong, or useless!
>
> > **Comment 6:** It appears novel because it uses neural networks instead of GPs as meta-learners.
>
> We apologise for not making this clearer, but we think our paper is novel beyond just switching the GPs in Li et al. for neural networks. In fact, our work and Li et al. solve fundamentally different problems.
>
> Our work takes a learner, and finds the non-trivial functions that the learner finds most easy to generalise after learning from a small train dataset labelled according to the function.
>
> Li et al. summarise sets of fixed neural networks via a GP.
>
> As such, Li et al. are describing the behaviour of sets of fixed neural networks. We are describing the way in which a particular network chooses to generalise. These are fundamentally different problems. We think our approach has more general utility in understanding how circuit design affects not just how a fixed set of neural networks behaves, but how they learn and choose to generalise.
>
> We changed the discussion of Li et al. to try and highlight this, including the following sentence:
>
> *In our work we do something very different* [to Li et al.]*: rather than focusing on a fixed, static set of networks, we find the inductive biases of learning algorithms via meta-learning easily learnt functions.*
>
> In fact, we only included this description because the title of Li et al.’s work (Meta-learning inductive biases of learning systems with gaussian processes) is very (very!) similar to ours, and we didn’t want people to think they tackled the same problems! It appears we failed in this goal, but does the new version of the manuscript achieve this?
>
> > **Effect of Meta-Learner Choice**
> **Weakness 3:** How does the meta-learner's choice affect the implicit bias (given the meta-learners have their own inductive biases)?
>
> > **Comment 1:** Although the paper offers an interesting approach, one important objection is that, as opposed to the paper from Li et al. 2021, the meta-learners would themselves require a meta-learner to understand their own biases.
>
> > **Comment 5:** Still, the resulting interpretability is lost because the meta-learners inductive bias is hard to understand.
>
> This is an insightful comment: indeed, the meta-learner has its own inductive biases. However, the role of the meta-learner and learner are asymmetric, so the inductive biases of the two networks play very different roles.
>
> With a finite set of data the learner is able to learn a limited space of functions, and it finds some of those functions easier to learn than others - this is the inductive bias. Through training the meta-learner explores this space of functions to find the one that is easiest-to-generalise. As long as (1) the function space of the meta-learner is large enough to include all the functions the learner is able to learn, and (2) the optimisation technique is good enough for the meta-learner to effectively search in this function space for the easiest-to-generalise function, then the particular shape of the meta-learner’s inductive bias is less important.
>
> Of course, like much neural network work, there is no hard theory behind this. In practice we tend to choose the meta-learner’s architecture to be a larger version of the learner’s, the motivation being that we want the space of learnable functions for the meta-learner to be a more expressive version of the learner’s, since the meta-learner only needs to learn to fit functions that the learner can generalise. For the set of meta-learners we have chosen (using the heuristics mentioned above) we have got consistent results. Of course, not all choices work: making the meta-learner linear, for example, would not work as it precludes learning non-linear functions.
>
> We would also like to point out that, as in our previous comment about the relationship between our work and Li et al.’s, the step from Li et al.’s work to ours is not just swapping GPs for neural networks, and hence losing interpretability. We are tackling a different problem to that in Li et al.

---

> > ### Author Response · Authors · 2022-11-18
> > **Extra comment on the choice of meta-learner's architecture**
> >
> > In response to another reviewer who had the same concern, we've added this comment to the main paper about choosing the meta-learner's architecture:
> >
> > *Our meta-learner must fit a function that the learner can generalise. To enable the meta-learner to learn all functions the learner might plausibly generalise well, its function class could usefully be a superset of the learner's. Therefore, we choose the meta-learner's architecture to be a slightly larger version of the learner's (though, beyond this, our findings appear robust, Appendix D).*
> >
> > And we've added a small Appendix D where we show that, for meta-learning on a simple feedforward ReLU network, the choice of how many layers in a feedfoward meta-learner doesn't matter, as long as it isn't zero!
> >
> > Fingers crossed this, together with our previous comments, helps to answer your concerns!

---

> > > ### Comment · Reviewer_wT2V · 2022-11-24
> > > **Response to the authors reviews.**
> > >
> > > I thank the authors for taking the time to respond and try to address the concerns I have.
> > >
> > > I believe that the new manuscript has gained clarity and that some necessary improvements have been made to the paper.
> > >
> > > I still believe that the circularity of the reasoning hasn't been addressed, and I still do not understand how we can study a theory using itself when we lack an essential understanding of it. It is an exciting idea, but I am not sure that using "neural networks" and "gradient optimization" are the solution to understanding "neural networks" and "gradient optimization."
> > >
> > > Moreover, as also pointed out by the other reviewer, most of what is learned are low-frequency basis, which confirms the existing theory. The critical question is, can such models learn anything besides low-frequency models?
> > >
> > > I will need further discussion with other reviewers on the subject.
> > >
> > > Thank you again for your efforts.

---

> > > > ### Author Response · Authors · 2022-11-27
> > > > **Response to Reviewer's Response to Author's Response to Reviewer's Review**
> > > >
> > > > Thanks for your comments, to briefly respond! (because we couldn’t resist ;) )
> > > >
> > > > > Can such models learn anything besides low-frequency models?
> > > >
> > > > Yes! In our updated manuscript we study the inductive bias of a sparsely connected network and find it has biases towards the neuron basis, on top of a low-frequency bias.
> > > >
> > > > As explained to other reviewers, we chose a network that was both biological and had a ‘not just low-frequency’ bias to answer as many questions at once (this one network hopefully answers both ‘beyond low-frequency?’ and ‘not biologically useful?’ concerns). We could have also studied networks with frequency-band biases etc., we can’t see a reason the tool should start failing then, when it hasn’t so far. We can try it if you’d like?
> > > >
> > > > > I still believe that the circularity of the reasoning hasn't been addressed
> > > >
> > > > If we were proposing a general theory of how neural networks work, it could indeed be difficult to get around this circularity.  However, our goal was more modest; we just wanted to get a handle on the inductive bias of networks.  That we were able to do, and we believe our results will be of practical use to practitioners.
> > > >
> > > > Importantly, as far as we can tell, no reviewer has argued that our tool doesn’t work – both to get a handle on inductive bias, and to get insight into biological circuits (as we have shown in the updated manuscript).
> > > >
> > > > > I am not sure that using "neural networks" and "gradient optimization" are the solution to understanding "neural networks" and "gradient optimization."
> > > >
> > > > That we agree with, but again, our goal was more modest: although we would love to understand neural networks and gradient optimization, here we just wanted to provide insight into inductive bias. Something we think we succeeded in doing, at least partially.
> > > >
> > > > All the best,
> > > >
> > > > Authors

---

> > > > > ### Comment · Reviewer_wT2V · 2022-12-02
> > > > > **Response to Authors.**
> > > > >
> > > > > Thank you for taking the time to respond.
> > > > >
> > > > > You have certainly convinced me that this piece of work is of good quality and has some qualities. The remaining of my personal objections would not be reflected by a score of 3. I believe that this piece of work deserves a score of 5, and I will update my score.
> > > > >
> > > > > I would like to express to the authors that I am impressed by the amount of work they have done.
> > > > >
> > > > > Best.

---

> > > > > > ### Author Response · Authors · 2022-12-03
> > > > > > **Thanks!**
> > > > > >
> > > > > > Thanks for your kind words, glad we could help! Anything else, just let us know!

---

> ### Author Response · Authors · 2022-11-18
> **Response (2/3)**
>
> > **Weakness 2:** The paper offers primarily qualitative comparisons (i.e., visual) rather than quantitative ones. Are there ways to properly understand the quality of the approximation of the implicit by different types of meta-learners?
>
> Thank you for highlighting this. Whenever there is a quantitative comparison available we agree it is valuable to do it. However, in general, it was not clear to us how to quantify the performance of the meta-learner. It finds functions that the learner finds easy-to-generalise. In the case of linear or kernel regression the shape of these functions are known allowing us to quantify the quality of the approximation. But in general there is no ground truth to compare to. We don’t know of other methods for extracting these functions!
>
> We hadn’t originally included quantitative comparisons for all the settings with the kernel learner, apologies, we’ve now fixed that. Figure 3 now includes a quantitative comparison to the predicted spherical harmonics.
>
> On the other hand, we can compare the quality of different meta-learners through the generalisation error of the functions they find. We haven't found significant differences among the range of meta-learners that we have used.
>
> > **Weakness 1:** The choice of the Sinkhorn distance seems somewhat arbitrary. How does the selection of the different objectives affect learning?
>
> You are right, the choice of Sinkhorn distance was arbitrary. We just needed a loss that would force the label distribution to take a particular form, and we found a convenient differentiable implementation of the sinkhorn divergence that worked well.
>
> We could definitely have used other regularizers. For example, in our new MNIST figure we use a loss that measures the variance of the meta-learner’s labels, bounded between 0 and 1. Notably, we cannot use the KL divergence as it is infinite if the support of the two distributions is disjoint. Since this is always the case at the beginning of training we had to search for alternatives.
>
> We tested a couple of other divergence metrics on the kernel regression example from figure 2: the maximum mean discrepancy from Gretton et al. 2012, and an energy statistic from Szekely et al. 2013, and found that the results were consistent. We have included these explorations in Appendix B. Certainly, the choice of divergence metric will have some effect on the learning dynamics, but the solution the meta-learner finds appears stable to these choices.
>
> > **Comment 3:** The authors rightfully point out that some tools are lacking to properly study their model.
>
> As you say, to extract meaning from our model in high dimensions some way to understand the functions implemented by the meta-learner is needed. This is, in general, hard.
>
> However, the updated manuscript now displays a series of insights derived either from visualisable low-dimensional systems. We have shown how our tool can provide insight for low-dimensional inputs (Fig. 2 - 5), by comparing to ground truth labels (Fig. 6), or by projecting the learnt functions onto an appropriate basis (Fig. 7). This we present as promising evidence that in many cases our tool can still provide insight.
>
> We will continue thinking about approaches to understand the inductive bias in various high-dimensional cases. However, we think the approach as it stands still has value that makes it worth sharing.

---

> ### Author Response · Authors · 2022-11-18
> **Response (3/3)**
>
> > **Not Even a Proof of Concept.
> Weakness 4:** What type of novel, non-trivial, insights can we obtain from the current paper? Various future directions and discussions are nicely laid out. However, as of now without them, this paper does not offer enough to be even a proof of concept.
>
> > **Comment 2:** Some of the claims of the usefulness of the approach are lacking as mostly known results are recovered, and little to no predictions are made that would have to be validated.
>
> > **Comment 4:** The paper, nonetheless, does not address critical questions and make enough predictions to make this paper even a proof of concept.
>
> Thank you for your feedback. Our understanding of a ‘proof of concept’ is that it should show the proposed method is feasible, but not necessarily use the method to break new ground. We believe our work demonstrates that the proposed method works, and hence is a proof of concept. Do you have lingering concerns about whether our proposed method works that we could help to answer?
>
> Now, we agree that claims about the usefulness of methods are more compelling with evidence, and that the previous version of our manuscript was relatively limited in this area. We have sought to fix this by introducing an additional biological example that uses the tool to derive non-trivial insights (at least, they were non-trivial to us!) into the effect of different connectivity structures on the inductive bias of a model of a particular biological circuit: the fly mushroom body.  A big recent trend in neuroscience is the availability of large amounts of connectomic data (i.e. which neurons connect to each other), however methods to interpret this data are few and far between. Our tool seems well-suited to this setting, explaining connectivity normatively via its effect on the inductive bias, and we’re excited to explore this direction further.
>
> To be specific, we built a model of the fly mushroom body with sparse connectivity. We show that (A) sparse connectivity aligns the easy-to-generalise functions with the input neuron basis, (B) if some input neurons have more connections than others then the learner finds it easier to generalise functions that depend predominantly on the highly connected inputs, (C) connectivity groupings (groups of input neurons for which, if you are connected to one, you are likely connected to others), lead to interesting generalisation biases when weights are constrained to be positive, specifically all members of the group are treated as influencing the output in the same way. These developments are detailed in section 5.3 and the new figure 7.
>
> Here, the tool seems to provide non-trivial insight into a current research problem in neuroscience. Does this answer your concern about the utility of our approach?

---

### Official Review · Reviewer_VQde · 2022-10-25

**Confidence:** 4
**Correctness:** 4
**Technical Novelty And Significance:** 4
**Empirical Novelty And Significance:** 3
**Recommendation:** 6

**Clarity, Quality, Novelty And Reproducibility:**

Overall, the paper is of high quality, introduces a novel idea, and is clearly written.

**Strength And Weaknesses:**

Strengths:
- Novel idea, well motivated
- Clear explanations and plenty of simple examples to build intuition
- The approach behaves reasonably for simple problems

Weaknesses
- The method is really only applied to simple networks. Why not apply the method to visualize or understand the inductive biases of more complex networks (even stuff like networks trained on MNIST—which is still a relatively easy task). Is it because the meta-learning fails for these bigger networks, or are the resulting learned labels hard to interpret?
- Most (all?) of the examples in the paper are cases where the learning algorithm has a bias towards smooth functions. It would be nice to see an example where the learning algorithm had some other kind of bias, and to show that the method recovers that specific inductive bias. You could even use simple 1D regression problems (instead of 2D) for simplicity. For example, the original MAML paper has a nice toy problem where they use MAML to meta-learn a network that has an inductive bias towards sinusoidal functions (by meta-training on sinusoids). If one were to take that network, and try to recover it's inductive bias using this method, could you recover what kind of function class was used to meta-train the network in the first place? I think that would be another powerful example of the technique (to demonstrate that the meta-learner can uncover structure besides "smooth" data).
- Instead of the orthogonalization procedure, what what happen if you just initialized using multiple random seeds and re-ran the meta-learning procedure? Would different random seeds solve the problem that the orthogonalization is trying to address?

**Summary Of The Paper:**

This paper proposes a simple method for visualizing the inductive bias of a supervised learning algorithm. The method involves meta-learning the labels for a dataset, such that the learning algorithm is able to easily generalize when trained on a subset of those labels. For example, if the inputs are 1D, the meta-learned labels will map out a function over the 1D space that corresponds to the "easiest" function for the learning algorithm to learn, hence one can interpret that function as representative of the inductive bias of the algorithm. The paper applies their method to a couple of toy examples (linear and kernel regression in 2D) to build intuition, as well as to a feedforward ReLU network and a spiking neural network.

**Summary Of The Review:**

Overall, I found this approach pretty interesting as a way to understand the function of a network. I think I would need to see more examples of being able to visualize or understand the inductive bias of larger-scale networks, or at the very least more toy examples where the inductive bias was something deeper beyond "smooth functions are easier to model". If this concern is addressed I would consider increasing my score.

---

> ### Author Response · Authors · 2022-11-18
> **Thank you for the review! Response (1/2)**
>
> We thank you for your detailed and useful comments on our work, it is much appreciated! We have introduced new results accordingly, and hope that these changes, along with our response here, address your concerns. We look forward to continued discussion in the coming weeks.
>
> > **Weakness 2: Only Smooth Biases.** Most (all?) of the examples in the paper are cases where the learning algorithm has a bias towards smooth functions. It would be nice to see an example where the learning algorithm had some other kind of bias, and to show that the method recovers that specific inductive bias. You could even use simple 1D regression problems (instead of 2D) for simplicity. For example, the original MAML paper has a nice toy problem where they use MAML to meta-learn a network that has an inductive bias towards sinusoidal functions (by meta-training on sinusoids). If one were to take that network, and try to recover it's inductive bias using this method, could you recover what kind of function class was used to meta-train the network in the first place? I think that would be another powerful example of the technique (to demonstrate that the meta-learner can uncover structure besides "smooth" data).
>
> > **And Comment 2:** I think I would need to see more toy examples where the inductive bias was something deeper beyond "smooth functions are easier to model"
>
> This is a legitimate concern. We have sought to address this by including a new biological example that displays inductive biases beyond purely smoothness (your MAML suggestion was very good, we appreciate the suggestion, and if we had time and space we would have included it. However, since other reviewers wanted an example of extracting biological insight, we decided to kill two birds with one stone and only study the biological example).
>
> Our biological example studies the effect of various observed connectivity patterns on the inductive bias of a biological pattern classifier: the fly mushroom body. We built a model of the fly mushroom body with sparse connectivity. We show that (A) sparse connectivity aligns the easy-to-generalise functions with the input neuron basis, (B) if some input neurons have more connections than others then the learner finds it easier to generalise functions that depend predominantly on the highly connected inputs, (C) connectivity groupings (groups of input neurons for which, if you are connected to one, you are likely connected to others), lead to interesting generalisation biases when weights are constrained to be positive, specifically all members of the group are treated as influencing the output in the same way. These developments are detailed in section 5.3 and the new figure 7.
>
> These seem like fairly non-trivial biases, beyond simply smoothness, that our scheme has successfully extracted. Do they address your concerns?
>
> > **Weakness 1: Only Simple Networks.** The method is really only applied to simple networks. Why not apply the method to visualize or understand the inductive biases of more complex networks (even stuff like networks trained on MNIST—which is still a relatively easy task). Is it because the meta-learning fails for these bigger networks, or are the resulting learned labels hard to interpret?
>
> > **And Comment 1:** I think I would need to see more examples of being able to visualize or understand the inductive bias of larger-scale networks
>
> This is again, a very legitimate concern. We took your suggestion and developed an MNIST example. We used the MNIST 0 and 1 digits as our input dataset and asked the meta-learner to assign a label to each digit, bounded between 0 and 1, that had high variance but which a convolutional neural network learner found easy to generalise. The meta-learner learnt to assign one extreme label (either 0 or 1) to the 0 digits, and the other extreme to the 1s, i.e. in searching for the most generalisable classification it rediscovered the MNIST digits! This is detailed in section 5.2 and figure 6.
>
> This does not address the deeper concern that interpreting these inductive biases is hard in high-dimensions. However, we now have two examples of this extraction process on input spaces more than 3 dimensional. First, on MNIST where we find the meta-learner simply aligns with the MNIST labels. Second, in the biological example where, by projecting the function onto appropriate subspaces, we were able to understand the inductive bias over a 5-dimensional input space (and we could have easily extended these techniques to higher dimensional inputs). Neither of these approaches is a general solution to what is a hard problem: understanding high-dimensional inductive biases. We hope that they point to the kind of approaches that could be used, and provide hope that, while it is difficult, understanding these high-dimensional biases is possible!

---

> > ### Comment · Reviewer_VQde · 2022-12-01
> > **re: additional experiments**
> >
> > Thanks for the additional experiments, they are informative.
> >
> > For the spiking network, correct me if I'm misunderstanding, but it still seems to me like the inductive bias is smoothness as a function of phase? Can we distinguish between "low frequency functions" and "smooth functions"?
> >
> > For MNIST, it is comforting to know that the method learns to correctly label the digits, but that doesn't shed much light on the inductive biases of the learner. To really get at that, it seems to me like you would want to feed in synthetic input images that were explicitly designed to tease apart inductive biases.
> >
> > This is a bit of a separate line of thinking, but after seeing these new results I am wondering if you could also, instead of starting the learner off with randomly initialized weights, start the learner using a checkpoint after pre-training on a separate task (such as MNIST). Would that shed light on the properties that a particular network has learned? (as opposed to the inductive bias of the architecture).

---

> > > ### Author Response · Authors · 2022-12-02
> > > **Response to Additional Experiment Comments**
> > >
> > > Hi! Thanks for the comments!
> > >
> > > > For the spiking network, correct me if I'm misunderstanding, but it still seems to me like the inductive bias is smoothness as a function of phase?
> > >
> > > Absolutely, for the spiking network (figure 5) there is only a bias towards smoothness as a function of phase.
> > >
> > > > Can we distinguish between "low frequency functions" and "smooth functions"?
> > >
> > > We don’t think so, or at least that wasn’t what we were trying to do!
> > >
> > > We’re not sure this is answering your question, but perhaps we are misunderstanding each other: we were not pointing to the spiking network in our response. We were trying to show that our tool can find ‘not just smoothness biases’ (or ‘not just low frequency’). To this end, we were pointing to figure 7 in our new submission, which is a rate network model of the fly mushroom body. This shows how sparse connectivity biases the network towards labellings that align with the neuron axes. These functions are still smooth, but this goes beyond simply ‘smooth functions’, or ‘low frequency’, no?
> > >
> > > Let me know if we’ve misunderstood something in this comment.
> > >
> > > > For MNIST, it is comforting to know that the method learns to correctly label the digits, but that doesn't shed much light on the inductive biases of the learner. To really get at that, it seems to me like you would want to feed in synthetic input images that were explicitly designed to tease apart inductive biases.
> > >
> > > We agree. This example was designed to answer a simple question: does our method work on high dimensional input data, which it does. Certainly, it was not very enlightening about the inductive bias of the network, for which we agree something beyond MNIST would be needed. But to reiterate, our previous manuscript left two possibilities open: either our method failed on high-dimensional inputs, or interpreting the inductive bias was hard in high-dimensions. We were trying to show it was the latter.
> > >
> > > Now, improving our ability to understand the high-dimensional inductive bias is an important direction. We quite like the idea of generating the inputs and the outputs from the meta-learner (as mentioned in the discussion), which may be a way to choose the synthetic inputs, but this is something we’re still working on.
> > >
> > > > This is a bit of a separate line of thinking, but after seeing these new results I am wondering if you could also, instead of starting the learner off with randomly initialized weights, start the learner using a checkpoint after pre-training on a separate task (such as MNIST). Would that shed light on the properties that a particular network has learned? (as opposed to the inductive bias of the architecture).
> > >
> > > That’s a good idea, I think that would be very interesting, and I believe is related to your earlier suggestion about finding the inductive bias of a MAML-trained network. This would actually be a super-fun way to find how training on one task changes the behaviour of the network on downstream tasks. We should try it! (or you should! With our handy, easy-to-use, github repo!) Thank you for the suggestion! (Though it’s too late to try this and put it in our current submission)
> > >
> > > Thanks for engaging, and happy to chat more!

---

> > > > ### Author Response · Authors · 2022-12-08
> > > > **Happy about concerns?**
> > > >
> > > > Hi!
> > > > Just to check before the deadline closes, did this answer your concerns?! There was potentially some confusion on our side.

---

> ### Author Response · Authors · 2022-11-18
> **Response (2/2)**
>
> > **Weakness 3: Is Orthogonalization Necessary?** Instead of the orthogonalization procedure, what what happen if you just initialized using multiple random seeds and re-ran the meta-learning procedure? Would different random seeds solve the problem that the orthogonalization is trying to address?
>
> This is a smart comment. Indeed, if there is a space of functions that the learner finds roughly equally easy to learn then re-running the meta-learner with different seeds explores this part of function space. For example, in the kernel regression problem described in section 2 there are 3 first-order spherical harmonics that are equally easy to learn. Re-running the meta-learner without any orthogonality constraint leads the meta-learner to meta-learn a different function at each run that lies somewhere in this 3-dimensional space.
>
> However, there are additional functions that the learner can still generalise, just not as well. For example, the higher order spherical harmonics in figure 3. We need the orthogonalisation to force the meta-learner to tell us about these functions. Just by re-running it has never yet produced a second-order spherical harmonic. This is in fact telling us that our meta-learner optimisation is working. The second-spherical harmonics are harder to generalise, so a well optimised meta-learner should minimise the generalisation error by choosing only first-order spherical harmonics, as we observe.
>
> We include this discussion and an associated figure in appendix C.
>
> There are likely other approaches beyond orthogonality that achieve the diversity in meta-learners that we are searching for, but orthogonality was simple and effective, and in the linear and kernel setting it truly provides a complete description of the inductive bias.

---

> > ### Comment · Reviewer_VQde · 2022-12-01
> > **re: orthogonalization**
> >
> > Makes sense, thanks

---

### Official Review · Reviewer_QUHj · 2022-10-26

**Confidence:** 4
**Correctness:** 3
**Technical Novelty And Significance:** 3
**Empirical Novelty And Significance:** 3
**Recommendation:** 6

**Clarity, Quality, Novelty And Reproducibility:**

The writing is generally clear and easy to understand, except in the discussions. The order of points while describing the issues with practical applications seems to have been reversed later in the discussions. Furthermore, it is not very clear why access to the gradients of the learning system is required.
The proposal seems to have merit and seems to be a promising direction in general. It is also a novel framework (to my knowledge). But given the lack of theoretical backing and empirical evidence in challenging problems in high-dimensions, I am unsure how this method scales with task (and network) complexity.
The authors plan to release their code on github, which should improve the overall reproducibility of the work.

**Strength And Weaknesses:**

Strengths:
1. The paper is very well-written (barring a few typos) and it is easy to follow and understand the core tenets of the proposed framework.
2. The authors present incremental evidence to test out different hypothesis pertaining to their proposal. In doing so, they enable the reader to develop the intuitions behind what the expect from the proposed framework, when put in practice.
3. The idea seems simple and elegant, yet powerful and highly applicable to systems identification problems in both the fields of machine learning and neuroscience.

Weakenesses:
1. A key weakness of the work is its limited empirical validation or experimental evidence, specifically for non-toy problems. Although the intuitions presented in toy tasks are clear and illustrate the usefulness of the method, it remains to be seen how well the bootstrapped learning framework performs in high dimensions.
2. A considerable failure mode in high-dimensional bootstrapped learning, something that this work proposes with the use of a learner that uses outputs of the meta-learner and vice versa, is dimension collapse. Specifically, it has been shown (for self-supervised bootstrapped learning) that often the functions learned are low-rank and therefore, the outputs do not span the entire space. Owing to this possible failure mode, it is reasonable to wonder whether this framework would be as applicable in high-dimensions as it is for low dimensionality problems.
3. I was a bit confused about the utility of the proposed method for biological circuits, specifically how does the proposal fit in when attempting to understand the inductive bias of biological circuits. Firstly, the framework expects the learner to be given some output. How would such a training protocol look like in a particular circuit in a behaving animal? It would be nice to clarify this in the main text as a major motivation revolves around using the framework for biological circuits. Secondly, is it necessary that the system requires gradients to learn? In my understanding, the framework would work if there is some learning happening in the learner. Or could the framework still work without any plasticity in the learner? Have the authors tried testing the limits of their proposal by varying the learning rate for the learner.
4. The authors present their framework and demonstrate that it works for low-dimensional toy settings but it is not clear why or how it does so. It would be nice to have some intuitive explanation or insights from learning theory or dynamical systems that conveys how this meta-learning setup converges to the inductive bias functions of a learning system.

**Summary Of The Paper:**

The authors present a meta-learning framework to infer a system's or circuit's inductive bias. In their work, the authors claim that their method connects architectural design choices to function space features. This work demonstrates the usability of the method in inferring the inductive bias for relatively simple models, i.e. linear and kernel regression as well as for shallow neural networks and spiking neural networks. The authors also leverage their framework and coupled with an adversarial training strategy, they demonstrate a potential application of learning functions that distinguish the inductive bias of two models with different architectures. Although current experimental evidence is limited to low-dimensional toy problems, the proposed method seems promising in understanding the inductive bias enforced by different architectural choices in high-dimensional end-to-end learning.

**Summary Of The Review:**

Although the general idea seems exciting and promising, I have my doubts over the scalability of this method. Unless authors can provide more evidence from higher dimensional problems (maybe something with MNIST where you try to visualize canonical functions it learns over the pixel space), the utility of the method seems to be limited. Moreover, it is not very clear from the discussions how this method can be used in computational neuroscience. Therefore, my current assessment is marginally below the threshold but I am happy to revisit my rating if the authors address some of my concerns raised above.

---

> ### Author Response · Authors · 2022-11-18
> **Thanks for the review! Response (1/3)**
>
> We would like to thank you for carefully reading and commenting on our work. We appreciate your comments, which, as we detail below, we have tried to address, and which have helped us to improve the manuscript. We look forward to discussing any remaining concerns further.
>
> > **Questionable Utility for Computational Neuroscience. Comment  5:** Moreover, it is not very clear from the discussions how this method can be used in computational neuroscience.
>
> > **Weakness 3A:** I was a bit confused about the utility of the proposed method for biological circuits, specifically how does the proposal fit in when attempting to understand the inductive bias of biological circuits.
>
> We apologise for this lack of clarity. We have sought to show how our tool can be used in neuroscience in a new figure, figure 7, and the corresponding description in section 5.3. We use our method to understand recent connectomic data (i.e. which neurons connect to each other). This is a large major new source of data in neuroscience, but there aren’t yet many methods to interpret this data. We show that our tool provide a role for different patterns in the connectomic data through their effect on the inductive bias of the circuit.
>
> In particular, we build a model of the mushroom body, an olfactory circuit in the fruit fly. We show that: (A) sparse connectivity aligns the easy-to-generalise functions with the input neuron basis, (B) if some input neurons have more connections than others then the learner finds it easier to generalise functions that depend predominantly on the highly connected inputs, (C) connectivity groupings (groups of input neurons for which, if you are connected to one, you are likely connected to others), lead to interesting generalisation biases when weights are constrained to be positive; specifically all members of each group are treated as influencing the output in the same way.
>
> This has already given us neuroscientific insights, and we foresee many potential uses for this tool down the line.
>
> > **Weakness 3B:** Firstly, the framework expects the learner to be given some output. How would such a training protocol look like in a particular circuit in a behaving animal? It would be nice to clarify this in the main text as a major motivation revolves around using the framework for biological circuits.
>
> It is true that in general, the instances when the brain works by pure supervised learning, receiving labels from some external source, are relatively rare. However, many processes can be well approximated as performing supervised learning, it is a classic assumption in much theoretical neuroscience, and we think our tool could be useful wherever these approaches have gained traction.
>
> To highlight a few examples:
>
> 1. Many circuits receive error signals that describe how a particular input was misclassified, such as the cerebellum (Shadmehr, 2020) or olfactory system (Aso et al. 2014), which makes it natural to model the system as supervised.
> 2. One brain area might provide a supervised learning signal for another. For example, systems consolidation is a classic model of how fast learning in one brain area provides a supervised training signal for a second brain area. Examples include the classic systems consolidation where the hippocampus trains the cortex (McClelland et al. 1995, Sun et al. 2021), or a recent model where the cerebellum trains the motor cortex (Pemberton et al. 2022).
> 3. The brain contains many internal models of how the world operates, for example the famous grid cell internal model of 2D space (Hafting et al. 2005). Theories of how these models are trained typically assume that the internal model makes a prediction of the future, and this is later compared to observations, providing a supervised error signal to train your internal model (Whittington et al. 2020).
> 4. Finally, replay - in which previously experienced episodes are replayed during rest or sleep - is a common brain-wide phenomena whose functional role has often been modelled using supervised learning (van der Ven et al. 2020).
>
> There are likely more examples, but hopefully these highlight how we can study the brain via twists on supervised learning. Since our tool is applicable to all these settings (as well as explicitly supervised learning!), we hope it will be useful. We’ve included the following sentences on this in the discussion:
>
> *This* [modelling a neural circuit as supervised learning] *is often reasonable. Some circuits contain explicit supervision or error signals, like the fly mushroom body or the cerebellum (Shadmehr, 2020); and generally brain areas that make predictions, i.e. all internal models, can use their prediction errors as a learning signal. Alternatively, some circuits are well modelled as one area providing a supervisory signal for another, as in classic systems consolidation (McClelland et al. 1995), or receiving supervision from a past version of themselves through replay (van der Ven et al. 2020).*

---

> > ### Author Response · Authors · 2022-12-04
> > **Second Response**
> >
> > Hi!
> >
> > As the end of the discussion period is fast approaching, do let us know if you have any further comments or concerns to which we can respond. We'd be interested to hear if our comments and changes to the manuscript helped answer your questions, especially about the biological relevance of our approach and the larger set of examples now provided!

---

> ### Author Response · Authors · 2022-11-18
> **Response (2/3)**
>
> > **Weakness 3C:** Secondly, is it necessary that the system requires gradients to learn? In my understanding, the framework would work if there is some learning happening in the learner. Or could the framework still work without any plasticity in the learner? Have the authors tried testing the limits of their proposal by varying the learning rate for the learner.
>
> > **Comment  2:** Furthermore, it is not very clear why access to the gradients of the learning system is required.
>
> Sorry for the lack of clarity.
>
> We will first clarify why we need access to the gradients of the learning system. Our goal is to minimise the generalisation error of the learner by changing the meta-learner’s parameters. To do this we take gradients of the generalisation error:
>
> $\mathcal{L}_g = \sum_i (z_i - \hat{z}_i)^2$
>
> Where $z_i$ is the meta-learner’s label on datapoint $i$ and $\hat{z}_i$ the learner’s. The gradient of this has two terms. The first is simple, proportional to:
>
> $\frac{\partial z_i}{\partial \theta}$
>
> where $\theta$ are the meta-learner's weights. The second is more complex, proportional to:
>
> $\frac{\partial \hat{z}_i}{\partial \theta}$
>
> Since the learner assigns labels based on the training dataset that it received from the meta-learner, $\hat{z}_i$ are also functions of the meta-learner’s parameters. Thus, we also have to take the derivative of the learner’s labels with respect to the meta-learner’s labelling of training data. This requires taking gradients through the learning procedure.
>
> Importantly, this is an orthogonal question to how the learner learns. The learner could learn by gradient descent, or by more biologically plausible schemes such as Hebbian plasticity, or indeed not learn at all. We just need to be able to take gradients through the learning procedure with respect to the meta-learner’s parameter.
>
> Now, the case where the learner has no plasticity (and hence the name learner is a misnomer) is particularly easy, since this means the learner is performing a fixed input-output mapping. Its inductive bias is therefore simple: it is inductively biased towards one fixed mapping, and cannot do any other!
>
> For a fixed amount of training, varying the learner’s learning rate interpolates between the setting we have been considering thus far and the fixed case, before very large learning rates eventually causing the learner’s weights to explode. There is a fairly large range of learner’s learning rates for which we get reasonable results, though we haven’t yet quantified this range exactly. There are likely other interesting effects of changing the learning rate, but we haven’t found them in our explorations so far.
>
> > **Focus on Low-Dimensional Toy Problems**
> **Weakness 1:** A key weakness of the work is its limited empirical validation or experimental evidence, specifically for non-toy problems. Although the intuitions presented in toy tasks are clear and illustrate the usefulness of the method, it remains to be seen how well the bootstrapped learning framework performs in high dimensions.
>
> > **Comment  4:** Unless authors can provide more evidence from higher dimensional problems (maybe something with MNIST where you try to visualize canonical functions it learns over the pixel space), the utility of the method seems to be limited.
>
> This is a reasonable concern. We have taken your suggestion and developed an MNIST example. In section 5.2 and figure 6 we now take the 0 and 1 digits from MNIST and meta-learn a high entropy labelling (for labels bounded between 0 and 1) that a learner finds easy to generalise. The meta-learner rediscovers the MNIST digit classification, labelling the 0 digits with one extreme label (either 0 or 1) and the 1 digits with the other extreme.
>
> We consider this exciting – the method works for higher-dimensional inputs! And it shows that the most generalisable classification of two MNIST digits was to split them into the pre-existing MNIST categories, which makes sense.
>
> There is the remaining concern that interpreting the functions learnt by the meta-learners for higher dimensional inputs will be hard. We could not think of a way to represent the meta-learner’s functions in the MNIST example. It is, after all, a function on a 768-dimensional input space.
>
> Extracting insight from these meta-learners will probably have to be done in a problem-specific way. For example, in the previously mentioned biological example with sparse connectivity we could get a lot of insight by projecting onto particular subspaces. This gives us hope that in many cases of interest similar approaches can be used to derive insight from our tool.

---

> ### Author Response · Authors · 2022-11-18
> **Response (3/3)**
>
> > **Weakness 2:** A considerable failure mode in high-dimensional bootstrapped learning, something that this work proposes with the use of a learner that uses outputs of the meta-learner and vice versa, is dimension collapse. Specifically, it has been shown (for self-supervised bootstrapped learning) that often the functions learned are low-rank and therefore, the outputs do not span the entire space. Owing to this possible failure mode, it is reasonable to wonder whether this framework would be as applicable in high-dimensions as it is for low dimensionality problems.
>
> This is an interesting comment. Dimension collapse is, indeed, a fundamental problem for our approach (though we did not know it had a name). Even in our toy examples we are fighting against dimension collapse, since, of course, the easiest function for a learner to learn is the trivial one where all datapoints are given the same label. To avoid this we add a term to the loss that forces the distribution of the labels to take a particular form (e.g. uniform between -1 and 1) or have high entropy.
>
> Moving to high-dimensional input data does not make this problem any worse, as long as the output dimensionality remains low. As shown by our MNIST example, forcing the labels to have high entropy avoids dimension collapse, as in our earlier toy examples.
>
> More concerning would be a move to high-dimensional output systems (fortunately a less common case). In many settings it is likely novel regularisors would have to be defined to avoid dimension collapse, though this does not seem like an impossible task. If one has a reasonable idea what the output distribution should look like, then regularising the label distribution correspondingly using a divergence measure seems plausible.
>
> > **Weakness 4:** The authors present their framework and demonstrate that it works for low-dimensional toy settings but it is not clear why or how it does so. It would be nice to have some intuitive explanation or insights from learning theory or dynamical systems that conveys how this meta-learning setup converges to the inductive bias functions of a learning system.
>
> We agree with you, it would be great to have some insight into how the meta-learner converges. However, this seems like a known hard problem: we do not fully understand how neural networks do so well at minimising loss functions when trained by gradient descent. Since, from the perspective of our meta-learner, our loss is just another loss, we face the same hard problem as people trying to understand how neural networks behave in general. As such, we don’t think we have anything particularly valuable to offer. Did you have a specific line or type of work in mind that could be useful to us?
>
> > **Comment  1:** The writing is generally clear and easy to understand, except in the discussions. The order of points while describing the issues with practical applications seems to have been reversed later in the discussions.
>
> Thank you for the helpful comment. You’re right, the ordering was off, now fixed.
>
> > **Comment  3:** But given the lack of theoretical backing and empirical evidence in challenging problems in high-dimensions, I am unsure how this method scales with task (and network) complexity.
>
> Hopefully the two new examples we have added will persuade you that (A) this method continues to find easy-to-generalise functions for higher dimensional inputs (B) this method can provide significant insight into biological systems. Of course, the method will fail eventually, but we would argue that it is sufficiently rich to be useful in many problems.
> We look forward to discussing further during the coming weeks.

---

> ### Comment · Reviewer_QUHj · 2022-12-05
> **Reply to Authors' response**
>
> I would like to thank the reviewers for their detailed responses and making significant changes to the manuscript. I would also like to apologize for the delay in my response. Given the discussion I had with other reviewers and the AC, as well as the updates to the manuscript, I believe that this paper presents an interesting framework and deserves to be accepted to the conference for more discussions. Therefore, I have adjusted my score accordingly.
> Below, I add some thoughts on the authors' responses to my initial concerns:
> 1. Utility of the tool for understanding neural circuits: I appreciate the efforts put in by the authors to show that their tool can be applied to infer inductive bias from connectomic data. I must admit, I had misunderstood what the author meant when I read the initial submission and the example of the fly mushroom body has made it more clear how their method can be applied. If I understand correctly, the authors propose that their method can be used to understand the inductive bias of a neural network that shares the same connectivity profile as a neural circuit. In my understanding, several properties of a neural circuit can impact its inductive bias, with connectivity profile being one of them. I agree with the authors that their method helps understand the inductive bias due to the connectivity. It would be great if the authors could clarify this aspect in the abstract or introduction.
> 2. Applicability to supervised learning: I agree that the authors that supervised learning has been leveraged in other avenues of computational neuroscience and this assumption in itself is not a drawback or limitation of the method.
> 3. Dimensional collapse in high-dimensional learning: I am still not completely convinced that forcing a high entropy output distribution of labels can alleviate the problem of dimensional collapse. In my understanding, it helps with avoiding representation collapse but dimensional collapse could still occur and hyperparameter tuning becomes a crucial component. However, I would defer to the authors' experience with their method. Moreover, the MNIST example adds more weight to their argument that dimensionality collapse is not a significant concern.
> 4. Understanding of why the method works: This was not a major concern, to be honest. It is something that would be good to have but I agree that it is indeed a harder problem. A potential example of this sort of work could be [this paper](https://arxiv.org/abs/2102.06810) that looks at self-suprvised learning without negative pairs, ie bootstrapped learning settings -- which has some similarities to the learning setup presented here.
> 5. Thank you for responding to the rest of the points. The response as well as the updates to manuscript have clarified my concerns.
>
> Overall, I have suggested accepting the paper and have increased my score accordingly.

---

> > ### Author Response · Authors · 2022-12-05
> > **Response to Reply**
> >
> > Thank you for your response and appreciation for our work! A couple of brief comments:
> >
> > >  Point 1: If I understand correctly, the authors propose that their method can be used to understand the inductive bias of a neural network that shares the same connectivity profile as a neural circuit.
> >
> > Exactly!
> >
> > > It would be great if the authors could clarify this aspect in the abstract or introduction.
> >
> > Sorry for the confusion. In any future versions of the manuscript we will try to make it clearer how the inductive bias of the neural network is used as a model of a biological system.
> >
> > > Point 4: *Understanding why tool works.* A potential example of this sort of work could be this paper that looks at self-suprvised learning without negative pairs, ie bootstrapped learning settings -- which has some similarities to the learning setup presented here.
> >
> > I had not seen this paper, it looks very nice. I had not realised people had made such in-roads into understanding neural network phenomena with simple analytic models. I look forward to digging into this area more - thank you for the suggestion!
> >
> > Thanks for your time and comments, which have helped us improve the manuscript significantly!

---

### Author Response · Authors · 2022-11-18
**General Response to Reviewers**

We would like to thank the reviewers for their careful reading and helpful comments. We appreciate your feedback, which has prompted us to improve our work significantly (in our opinion!).

We have sent replies to each of you detailing our responses to your comments, and have made numerous edits to the manuscript in response. In our new submission we have highlighted larger changes in a sort of plum colour. Code to generate all figures will be posted to an anonymous git repo in the coming days. Below we summarise the major changes, and we look forward to ongoing discussion over the coming weeks!

**Summary of Major Changes**

There were legitimate concerns about the breadth of the examples presented in the paper, and how that reflected on the broader applicability of the method, e.g.:

1. All our examples were toy low-dimensional problems, does it work in higher dimensional spaces? (Reviewers QUHj & VQde)
2. It was unclear how they could be useful for computational neuroscience (Reviewer QUHj)
3. All the effects we saw were smoothness priors, can it find anything else? (Reviewer VQde)
4. No non-trivial insights (Reviewer wT2V)

To answer this set of concerns we include two new examples

1. biological example inspired by connectomic data from the fly mushroom body. We show we can use our tool to interpret connectomic findings through their effect on the circuit’s inductive bias.
This provides a biologically useful example (2. above), with priors beyond simply smoothness (3. above), non-trivial insights into a system (4. above) and application to a slightly higher dimensional space (1. above).
2. An MNIST example where we learn labellings of MNIST that a learner network is able to generalise.
This provides evidence that our tool works in high dimensions (1. above)

We think this significantly improves the paper, and we hope reviewers agree!

---

> ### Author Response · Authors · 2022-11-27
> **Anonymous Github Link for Code**
>
> Here is an anonymous github repo with our code:
>
> https://anonymous.4open.science/r/Meta_Inductive_Bias_ICLR-6449/
>
> We're still refactoring the code for figure 5 into a nice jupyter notebook in the style of the others, but it will be there soon!

---

### Decision · Program_Chairs · 2023-01-20

**Decision:**

Reject

**Justification For Why Not Higher Score:**

The current model is limited to examining the inductive biases of differentiable supervised algorithms, which means its applicability in neuroscience is quite limited. It also has no clear applications in machine learning.

**Justification For Why Not Lower Score:**

The paper is novel and well-written, and could be the basis for future research, which makes an accept decision potentially appropriate. However, the current limitations make this decision difficult.

**Metareview: Summary, Strengths And Weaknesses:**

This paper presents a meta-learning technique for uncovering the inductive biases in a learning neural circuit. The basic approach is to train a meta-learner that can be used to predict which new tasks the target neural circuit will be effective at learning, thus telling us about its inductive biases. The authors show that in some very specific circumstances where the ground-truth inductive biases are known this approach is successful in uncovering those inductive biases.

This paper was a borderline case and not an easy decision. Some of the reviewers were quite positive, but others felt that the current incarnation was far too limited to be of real use to neuroscientists. Post rebuttals, which did provide some specific use cases, a meeting of the reviewers and AC (see more below) it was agreed that the paper was novel but the use cases were indeed limited to specific types of supervised learning, and that this is a very limited slice of potential real use cases in neuroscience. After consideration of this meeting, and discussions between the AC and SAC, it was decided that these limitations are too severe to make this paper pass the high bar of acceptance at ICLR.

**Summary Of Ac-Reviewer Meeting:**

In the meeting of the AC and the reviewers the principal point of discussion was whether the limited nature of the approach was sufficiently concerning to warrant a rejection, despite the fact that the core idea is novel and the paper well-written. By the end of the discussion, and in considering some of the authors' responses that tried to partially address the concerns, the AC was leaning towards an accept decision. However, after more discussion with the SAC it was decided that these limitations are too severe and it should be rejected.